# Structural classification of boron nitride twisted bilayers and ab initio investigation of their stacking-dependent electronic structure

Sylvain Latil,[1] Hakim Amara,[2, 3] and Lorenzo Sponza[2, 4]

[1]*Université Paris-Saclay, CEA, CNRS, SPEC, 91191 Gif-sur-Yvette, France*
[2]*Université Paris-Saclay, ONERA, CNRS, Laboratoire d'étude des microstructures (LEM), 92322, Châtillon, France*
[3]*Université de Paris, Laboratoire Matériaux et Phénomènes Quantiques (MPQ), CNRS-UMR7162, 75013 Paris, France*
[4]*European Theoretical Spectroscopy Facility (ETSF), B-4000 Sart Tilman, Liège, Belgium*

(Dated: August 29, 2022)

Since the discovery of superconductive twisted bilayer graphene which initiated the field of twistronics, moiré systems have not ceased to exhibit fascinating properties. We demonstrate that in boron nitride twisted bilayers, for a given moiré periodicity, there are five different stackings which preserve the monolayer hexagonal symmetry (i.e. the invariance upon rotations of 120°) and not only two as always discussed in literature. We introduce some definitions and a nomenclature that identify unambiguously the twist angle and the stacking sequence of any hexagonal bilayer with order-3 rotation symmetry. Moreover, we employ density functional theory to study the evolution of the band structure as a function of the twist angle for each of the five stacking sequences of boron nitride bilayers. We show that the gap is indirect at any angle and in any stacking, and identify features that are conserved within the same stacking sequence irrespective of the angle of twist.

Initiated by twisted bilayer graphene, moiré systems formed of 2D atomic layers have recently been established as a unique playground for highlighting novel and fascinating properties [1]. A tiny twist between the two van der Waals atomic layers can modify deeply their electronic properties as a consequence of the flattening of the band dispersion. In graphene, a flat moiré mini-band appears at specific "magic angles" [2, 3] whose occupation drives superconductive/insulating transitions which open new perspectives on the investigation of strong correlation in 2D systems [4–6]. In gapped twisted bilayers (e.g. semiconducting transition metal dichalcogenides) the moiré bands have an impact on the optical properties. For instance, by varying the twist angle it is possible to modulate the exciton lifetime [7], or the energy and intensity of emitted light [8–11]. In these systems, flat bands give rise to intriguing phenomena without the need of being twisted by specific "magic" angles [12–14].

Hexagonal boron nitride (hBN) is a cardinal compound in 2D materials research. Used mostly as incapsulating layer, it has nonetheless attracting properties on its own respect, mainly because of its large band gap ($> 6$ eV) [15, 16] which is at the origin of a strong UV emission [17, 18], single photon emission [19–24] and its application as gating layer in 2D electronics [25–28]. Recently ferroelectricity has been enabled in twisted hBN bilayers, thus expanding further its range of applications [29, 30]. In the bulk phase and in thin layers its optical properties are driven by excitons [31]. In hBN moiré systems, Lee and coworkers [32] observed an increase of the luminescence intensity and a decrease of the sub-band gapwidth for increasing twist angles. From the standpoint of atomistic simulations, geometries with small rotation angles require very large periodic cells (order of thousands of atoms) which are out of reach for most self-consistent numerical approaches [33]. As

for graphene [2, 34], tight-binding or continuous models based on the $k \cdot p$ approximation are more adapted to deal with very large systems and have therefore been developed [6, 14, 35, 36]. However these studies are incomplete on two aspects. First, the very nature of the band gap is still not elucidated while it obviously rules the optical and excitonic properties of monolayer and bulk hBN [15, 16, 37]. Second, the stacking sequence in bilayers is seldom considered and, when it has been, only two geometries were taken into account [33, 36]. Yet, it has been shown that the stacking sequence strongly influences the character of the gap [31, 38, 39] through long range interplanar interactions.

In this Letter, we investigate the electronic structure of twisted hBN bilayers by taking into account fully and on the same footing its dependence on the twist angle and the stacking sequence. As a first step, we demonstrate the existence of five and only five different stacking possibilities to construct hBN bilayers with hexagonal symmetry and provide a non-ambiguous nomenclature applicable to untwisted configurations as well and to any other homobilayer formed of hexagonal 2D materials. Stemming from this symmetry analysis, we employed density functional theory (DFT) to investigate the evolution of the band structure as a function of the twist angle for each of the five stackings.

To construct a tiling of rotated bilayers preserving long range translational symmetries, we first define coincident supercells [40]. Let us take a honeycomb lattice with primitive vectors $\mathbf{a}_1$ and $\mathbf{a}_2$ forming an angle of 60° and with the two atoms of the cell separated by $\boldsymbol{\tau}$. Then we define the $(q, p)$ hexagonal supercell as resulting from the vectors $\mathbf{A}_i^{(q,p)} = \sum_j M_{ij}^{(q,p)} \mathbf{a}_j$ defined by means of the matrix

$$M^{(q,p)} = \begin{bmatrix} q & p \\ -p & p+q \end{bmatrix} . \tag{1}$$

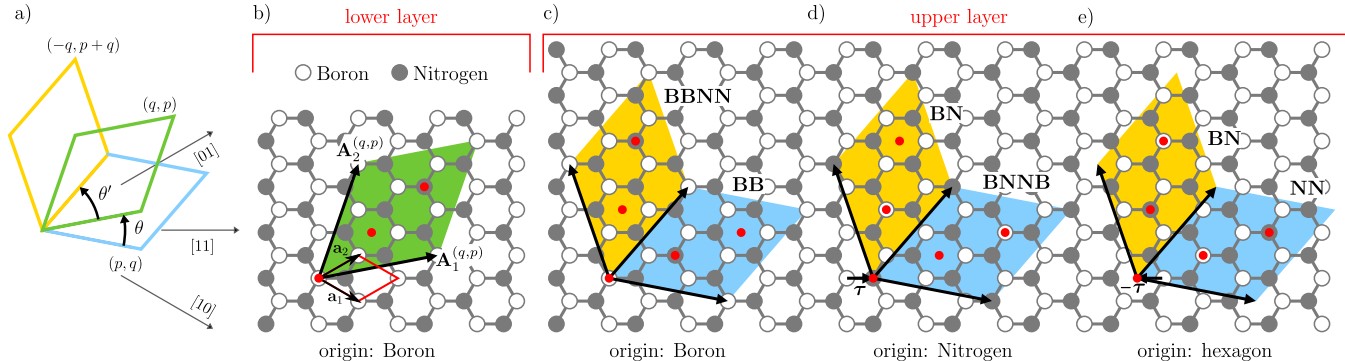

FIG. 1. a) Graphical representation of $\theta$ and $\theta'$ angles according to the $\{p,q\}$ integers. b) The lower layer supercell is $(q,p)_{\mathrm{B}}$. c) d) and e) The supercells of the upper layer $(p,q)_{\mathrm{X}}$ with X = B, N or H respectively are drawn in blue, and the corresponding $(-q,p+q)_{\mathrm{X}}$ supercells in yellow. High symmetry points are reported as red dots. In the examples $p=2$ and $q=1$.

Similarly we use equation (1) to introduce the $(p,q)$ and the $(-q,p+q)$ supercells. The resulting twist angles are given respectively by the formulae:

$$\tan\theta = \frac{\sqrt{3}(p^2-q^2)}{p^2+q^2+4pq} \quad \text{or} \quad \tan\theta' = \frac{\sqrt{3}(q^2+2pq)}{2p^2-q^2+2pq}.$$

The supercells defined above and the resulting twist angles are sketched in Figure 1.a.

The $p$ and $q$ integers obey to some constraints: they must be different and non zero, otherwise they lead to twist angles of 0° or 60°, they must have no common divisor, and the case $p-q$ multiple of 3 has to be excluded as it corresponds to non-primitive moiré supercells. Moreover, since twist angles are defined modulo 60°, the definition of the $M^{(\alpha,\beta)}$ matrices are not unique. We will then restrict ourselves arbitrarily to cases $p > q$ which imply that angles are positive and $\theta + \theta' = 60°$. Note finally that the notation introduced here for twisted bilayers can be employed also for untwisted structures taking $q = 0$ and $p = 1$.

Stacking the correct supercells is not enough to construct moiré hexagonal bilayers because the respective alignment is also crucial. Let us introduce a subscript labelling the origin of the supercell (B = boron, N = nitrogen, H = hexagon center). Without loss of generality we will always consider the supercell of the lower layer as being $(q,p)_{\mathrm{B}}$ (cfr. Figure 1b) while that of the upper layer can be any of $(p,q)_{\mathrm{B,N,H}}$ or $(-q,p+q)_{\mathrm{B,N,H}}$. As a consequence, one ends up with six bilayers listed in Table I and sketched in panels c), d) and e) of Figure 1 for the case $p = 2, q = 1$. In each supercell there are three direct-space high-symmetry points (red bullets in Figure 1): the points (0 0), (1/3 1/3) and (2/3 2/3) in the supercell reduced coordinates. Depending on the coincident atoms in these points, one can distinguish between (i) two geometries with a double sublattice coincidence per cell, the $(p,q)_{\mathrm{N}}$ and the $(-q,p+q)_{\mathrm{B}}$ ones, with a twist angle $-\theta'$, and (ii) the remaining four geometries with a single sublattice coincidence per cell and an an-

| upper layer | twist angle | symm. group | stacking sequence | double coincidence |
|---|---|---|---|---|
| $(p,q)_B$ | $+\theta$ | $p321$ | **BB** | no |
| $(p,q)_N$ | $-\theta'$ | $p321$ | **BNNB** | yes |
| $(p,q)_H$ | $+\theta$ | $p321$ | **NN** | no |
| $(-q,p+q)_B$ | $-\theta'$ | $p312$ | **BBNN** | yes |
| $(-q,p+q)_N$ | $+\theta$ | $p3$ | **BN** | no |
| $(-q,p+q)_H$ | $+\theta$ | $p3$ | **BN** | no |

TABLE I. The geometry of the five stackings of hBN twisted bilayers. The lower layer is based on the $(q,p)_B$ supercell.

gle of twist $\theta$. However it is trivial to demonstrate that the bilayers resulting from the stacking of $(-q,p+q)_{\mathrm{N,H}}$ on the $(q,p)_{\mathrm{B}}$ are related by a simple inversion and are therefore identical. All this boils down to five hexagonal stackings for the generic twisted hBN bilayer. As a consequence, we will designate univocally a twisted bilayer by the notation $STACK(q,p)$ where the $\{p,q\}$ pair relates to the supercell and hence the moiré periodicity and angles, and $STACK = $ BBNN, BNNB, BB, BN or NN relates to the atoms in the coincident sites. Images of these stackings, their layer symmetry group and the transformations to be applied to the upper layer to switch from one stacking to another (swapping of B/N atoms or translation by $\pm\tau$) are summarized in Figure 2 and Table I. It is worth recalling that with our conventions the angles are positive. Their sign comes from the chirality of twisted bilayers and is defined according to the screw angle separating B-N bonds at the atom-on-atom coincidence sites of the supercell, as depicted in the insets of the Figure 2.

For comparison, in the case of graphene bilayers both B and N labels become C, so the possible stackings are only two, but they have higher symmetry. The first belongs to the $p321$ layer group and to the odd bilayer graphene (BLG) set [34, 41–43], has a single sublattice vertical coincidence per cell and the twist angle is $\theta$. The second

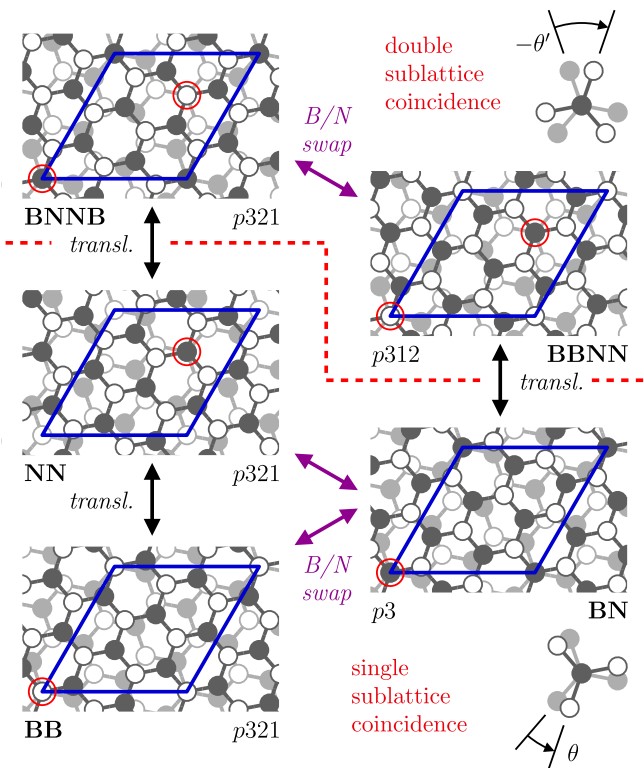

FIG. 2. The five stackings of hBN moiré structures, with $p = 2$ and $q = 1$. The sublattice coincidences are highlighted with red circles.

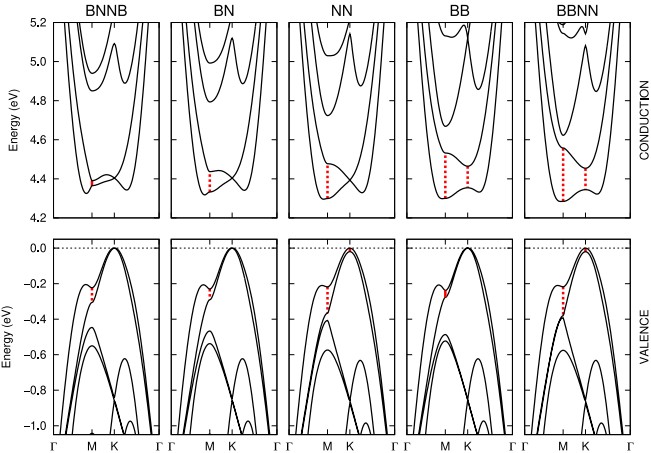

FIG. 3. Bottom conduction and top valence of the five principal stackings in the $(1, 2)$ supercell. Red vertical dashed lines highlight the notable splittings at M and K reported also in Table II.

belongs to the $p622$ layer group and to the even BLG set with hexagon-on-hexagon or double sublattice coincidence. Its rotation angle is $-\theta'$. Finally, if we swap the values of $p$ and $q$, we will obtain five new stackings which are the mirror images of the pristine structures. They will have the same electronic structure, and the twist angles will be $+\theta'$ for the BNNB and BBNN and $-\theta$ for the BB, BN and NN stackings. Complete definitions and demonstrations are given in Appendix.

Based on our robust symmetry analysis, we clearly identify five different stackings of hBN bilayers. Zhao and coworkers [33] studied two of them (the NN and the BN one) with a DFT method based on a tight-binding Hamiltonian and demonstrated that the stacking sequence has an impact on the spatial localization of the top valence and bottom conduction states. On the other hand, in a previous work [38] we proved that interlayer coupling, and so the stacking, is crucial in the formation of the indirect band gap of the bulk phase. These elements clearly indicate that a complete investigation involving all the stackings is mandatory. As a consequence, we have performed first-principle simulations with density functional theory (DFT) to investigate the impact of the stacking sequence on the band gap. We scrutinized thirty bilayers: six $\{p, q\}$ pairs per each stacking. All the pertinent calculation parameters can be found in Appendix F.

As a first step, we investigated the structural stability of the five principal untwisted bilayers and identified two main groups (see Figure 11 in Appendix G). In the three most stable structures (BN(0,1), BNNB(0,1) and BB(0,1)) the layers are separated by about 3.1 Å. The two least stable bilayers (BBNN(0,1) and NN(0,1)) are around 20 meV per formula unit at higher energy with larger equilibrium interlayer distances (around 3.4 Å). Regarding the electronic properties, untwisted bilayers with a boron-on-boron conicidence (BBNN(0,1) and BB(0,1)) have an indirect band gap whereas the other structures have a direct gap. More details about the untwisted bilayers can be found in Appendix G.

We now discuss twisted bilayers. We focus on the $(1, 2)$ configuration for all stackings because notable effects are more distinguishable. The DFT results are reported in Figure 3 inside the Brillouin zone of the supercell. It is important to recall that the preservation of the hexagonal symmetry of the supercell implies the conservation of their order-3 rotation axes without which the equivalence between the $K$ points of the Brillouin zone would be lost. Interestingly, our calculations reveal that the

| Structure | Top valence | | Bottom conduction | |
|---|---|---|---|---|
| | @M | @K | @M | @K |
| BNNB(1,2) | 83 | - | 25 | - |
| BN(1,2) | 61 | - | 104 | - |
| NN(1,2) | 148 | 20 | 178 | - |
| BB(1,2) | 38 | - | 232 | 110 |
| BBNN(1,2) | 163 | 20 | 273 | 110 |

TABLE II. The band splitting (meV) at M and K in the top valence and bottom conduction of the $(1,2)$ supercells. The symbol '-' indicates a band crossing. These features are highlighted with red vertical lines in Figure 3.

| family | $(q,p)$ cell | BNNB | BN | NN | BB | BBNN |
|---|---|---|---|---|---|---|
| $\delta = 1$ | (1,2) | 4.325 (71) | 4.318 (76) | 4.296 (88) | 4.299 ( 55) | 4.284 (60) |
| | (2,3) | 4.221 (30) | 4.217 (34) | 4.211 (38) | 4.203 ( 41) | 4.202 (42) |
| | (3,4) | 4.153 (15) | 4.153 (16) | 4.151 (17) | 4.145 ( 18) | 4.146 (19) |
| | (4,5) | 4.102 ( 5) | 4.103 ( 5) | 4.101 ( 5) | 4.098 ( 5) | 4.099 ( 5) |
| $\delta = 2$ | (1,3) | 4.284 (137) | 4.284 (137) | 4.284 (137) | 4.284 (136) | 4.284 (136) |
| | (3,5) | 4.240 ( 72) | 4.241 ( 72) | 4.240 ( 72) | 4.240 ( 72) | 4.241 ( 72) |

TABLE III. The DFT energy (eV) of the indirect band gap at different twist angles and stacking sequences. In parenthesis: energy difference between the direct and the indirect band gap in meV.

gap is always indirect irrespective of the stacking with values around 4.3 eV (see first row of Table III). By analyzing in details the electronic structure, we can distinguish the stackings according to characteristics at the $K$ and $M$ points. In the valence region we observe that when N atoms are on top of each other (the NN and the BBNN stackings), a band crossing is avoided in the top valence at $K$ while the splitting between the HOMO and HOMO-1 at $M$ is the largest. On the conduction band, the splitting between the LUMO and the LUMO+1 at $M$ is reduced along the sequence BBNN, BB, NN, BN and BNNB while the presence of B atoms on top of each other (BB and BBNN stackings) prevents a band crossing at $K$. All the features discussed here are highlighted with dashed vertical red lines in Figure 3 and reported in Table II. We expect these effects to be less important at extremal twist angles (i.e. close to 0° and 60°) because the immediate surroundings of each atom change progressively.

Let us now discuss the evolution of the band gap as a function of the twist angle. In Table III and in Figure 4 we summarize our DFT results on the indirect band gap and the difference between direct and indirect gap. First, we observe that the gapwidth gets smaller (higher) for smaller $\theta$ ($\theta'$), demonstrating a trend opposite to what predicted by continuous models [32]. Typically, for $\theta$ varying from 21.79° to 7.34°, the gap decreases by about 5%. Secondly we observe that in each stacking the gap remains indirect at all angles. This finding contrasts with density-functional tight-binding results where direct gaps at all twist angles are obtained instead [33]. A more detailed analysis reported in Appendix H allows us to affirm that it is not an artifact coming from $\sigma$ or nearly-free-electron states located at higher energies [15, 44–49]. We should stress that these results are reliable as long as one considers energy differences and trends, absolute gap energies being systematically underestimated by DFT. Indeed, we expect quasiparticle corrections, included for instance via the GW approximation, to be almost identical form one system to the other and to have minor effect on the dispersion of $s$ and $p$ states [15, 31], as demonstrated by the successful use of the scissor operator in BN compounds [16, 38, 39].

We can now pass to the investigation of the evolution of the full band structure as a function of the twist angle. In the main text we discuss two paradigmatic stackings, the BN and the NN and we report the corresponding twelve band structure plots in Figure 5. We refer the reader to the Appendix J for the other bandplots. We observe that conduction and valence bands get flatter at smaller $\theta$ (and larger $\theta'$) as highlighted in Figure 5. This implies the progressive creation of localized valence and conduction states in agreement with what shown by Zhao and coworkers [33]. For example, in the BN stacking at $\theta$ =7.34°, the HOMO and LUMO states are characterized by bandwidths around 0.09 eV and 0.16 eV, respectively. Flatter bands are not observed since this would demand much smaller angles which are inaccessible with our numerical resources. Because of the flattening of the bands, it is possible to tune the difference between indirect and direct gap through the twist angle, and so possibly to convert progressively the radiative decay pathway from a phonon-assisted emission to a direct recombination. This may have strong impact on the intensity of emitted light

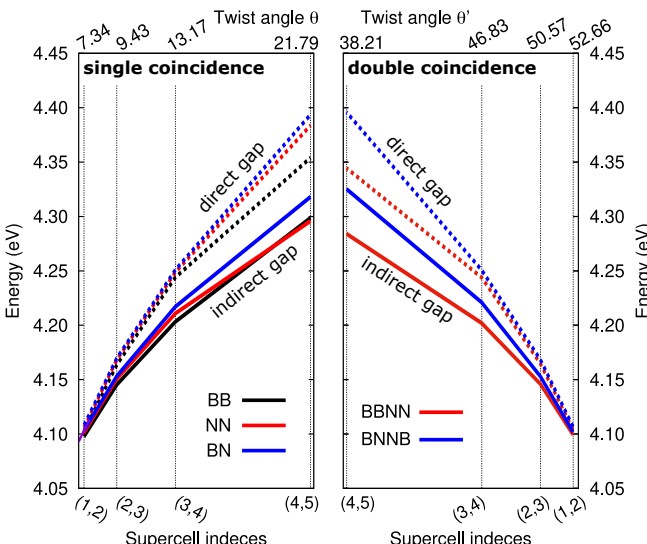

FIG. 4. Indirect gap (solid lines) and direct gap (dashed lines) of the five stackings as a function of the twist angle ($\theta$ or $\theta'$ depending on the stacking) within the $\delta = 1$ family.

(probability of recombination), its temperature dependence (through the coupling with phonons) and finally the life time of excitations.

In addition we observe that $\{p, q\}$ pairs can be grouped into families defined by the parameter $\delta = |p - q|$ that characterizes the interplay between crystalline structure (twist angle) and electronic structure (bands). In fact, the bands around the gap within the same family look similar but shrunk and flattened at small $\theta$ (or larger $\theta'$). Once more, the case $\delta$ multiple of 3 shall be excluded. Consider the family $\delta = 1$, corresponding to the first four plots from the left in the band plots of Figure 5. Here the valence bands present a maximum in $K$ and are formed of two bands dispersing almost parabolically, up to $M$ where one of the two deviates with a small bump. In conduction, two valleys are well discernible between $K$ and $\Gamma$ and around $M$, the latter forming the conduction band minimum. The last two plots from the left in the band plots of Figure 5 belong to the $\delta = 2$ family. These bandplots look very different from those of the other family, even though the gap remains indirect with the top valence at $K$. As before, one can see common features within this family despite the band shrinking. The valence band has a characteristic double-dome shape (with a dome on top of another) and a maximum in $K$. In the conduction band, the two bottom bands almost coincide in the $M - K$ path and present two minima close to or at $\Gamma$. We verified that the bottom conduction in the $\delta = 2$ family does fall in the $\Gamma - M$ high symmetry line (see Appendix I).

To conclude, we have demonstrated that in hBN bilayers there are five stackings that are invariant under rotations of $120°$ like the pristine hBN monolayers. We have listed the symmetry groups of these stackings, shown how to construct them and how to transform one into another and we have introduced a physically informative nomenclature allowing to identify them unambiguously. We also have provided a precise definition of the twist angle ($\theta$ or $\theta'$ depending on the stacking). All this contrasts with graphene bilayers, where only two stackings can be defined. Our nomenclature is completely general and can be applied to any homobilayer formed of hexagonal 2D materials (twisted as well as untwisted). Even though corrugation and domain relaxation have to be expected in experimental realization of these systems [30, 50, 51], these structural modifications will still be constrained by the stacking sequence. By performing DFT simulations, we have done a thorough study of the electronic structure of hBN bilayers taking into account both its dependence on the stacking sequence and the twist angle. In the first case, we have traced a correlation between the atom-on-atom coincidences and some characteristics of the states which form the gap. In the second case, we have shown that the gapwidth is always indirect irrespective of the twist angle and it decreases for decreasing $\theta$ or for increasing $\theta'$, differently from what previously predicted on the basis of less sophisticated simulation schemes [32]. Finally we have identified the structural parameter $\delta = |p - q|$ which allows to classify bilayers into families with similar band structures. The stacking- and angle-dependent properties discussed in this letter have special importance in possible twistronic applications. In fact these mechanisms are expected to have a strong impact on the optical properties of these bilayers and in particular on the direct manipulation of interlayer excitons which can be stabilized through the application of an external field.

The authors are thankful to Dr. F. Paleari for fruitful discussions and the dedicated analysis tools he provided. They also acknowledge the contribution of F. Ducastelle who seeded this work. Finally, they acknowledge funding from the European Union's Horizon 2020 research and innovation program under grand agreement N° 881603 (Graphene Flagship core 3) and from the French National Agency for Research (ANR) under the projects EXCIPLINT (Grant No. ANR-21-CE09-0016).

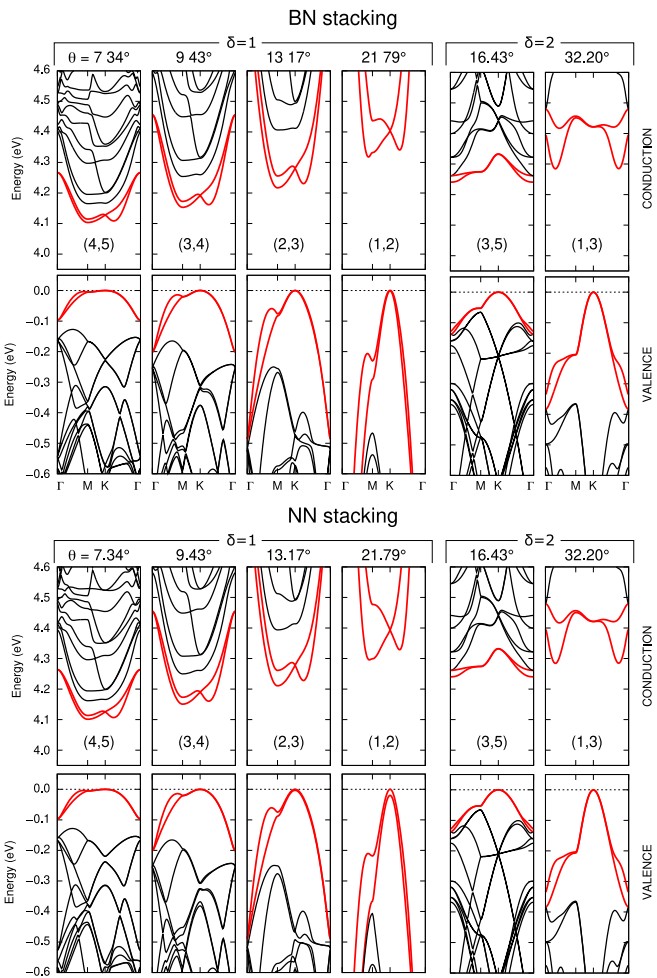

FIG. 5. Bottom conduction and top valence of the BN (top panel) and NN (bottom panel) stackings at different twist angles.

## APPENDICES

### A: Asymetric honeycomb supercells

As presented in the main article, we choose the two primitive vectors of the boron nitride monolayer $\mathbf{a}_1$ and $\mathbf{a}_2$ forming an angle of $60°$ and define the three vectors separating the nitrogen and the boron sublattices like:

$$\boldsymbol{\tau}_1 = +\mathbf{a}_1/3 + \mathbf{a}_2/3 \qquad \boldsymbol{\tau}_2 = \boldsymbol{\tau}_1 - \mathbf{a}_1 \qquad \boldsymbol{\tau}_3 = \boldsymbol{\tau}_1 - \mathbf{a}_2$$

A boron atom is located at the origin of the honeycomb and nitrogen is located at $\boldsymbol{\tau}_1$. A new periodic super-lattice is constructed with the new translational vectors $\mathbf{A}_1$ and $\mathbf{A}_2$ written on the basis $\{\mathbf{a}_1, \mathbf{a}_2\}$ like

$$\mathbf{A}_i = \sum_j M_{ij}\mathbf{a}_j. \tag{2}$$

In the bilayer system, the hexagonal supercell for the lower layer has been arbitrarily chosen as the one produced by the matrix

$$\mathrm{M}^{(q,p)} = \begin{bmatrix} q & p \\ -p & p+q \end{bmatrix} \tag{3}$$

and the upper layer is developed either with

$$\mathrm{M}^{(p,q)} = \begin{bmatrix} p & q \\ -q & p+q \end{bmatrix} \tag{4}$$

or with

$$\mathrm{M}^{(-q,p+q)} = \begin{bmatrix} -q & p+q \\ -p-q & p \end{bmatrix}. \tag{5}$$

In all these cases, $p$ and $q$ are integers. The vertical mirror planes along the $[1\,1]$ and $[1\,0]$ directions of the supercell are lost only if

$$p \neq 0, \ q \neq 0 \text{ and } p \neq q$$

then, we call such supercell *asymmetric*. These are the supercells considered in this work because they lead to twisted bilayers.

Lastly, the $\{p, q\}$ integers define also the parameter length, the surface $\Omega$ and the numer of atoms $\mathrm{N}_{\mathrm{at}}$ of the three supercells

$$|\mathbf{A}_i| = a\sqrt{p^2 + q^2 + pq} \tag{6}$$
$$\Omega = \Omega_0 \left(p^2 + q^2 + pq\right) \tag{7}$$
$$\mathrm{N}_{\mathrm{at}} = 2\left(p^2 + q^2 + pq\right) \tag{8}$$

where $\Omega_0 = \frac{a^2\sqrt{3}}{2}$ is the surface, and $a$ is the cell parameter of the honeycomb primitive cell.

As we mention in the main article, the origin of a generic $(k, s)$ supercell can be set either on an atom or on the center of a hexagon of the underlying honeycomb lattice. We want to analyze what happens at the direct-space high-symmetry points $(0\,0)$, $(\frac{1}{3}\,\frac{1}{3})$ and $(\frac{2}{3}\,\frac{2}{3})$ of the supercell where the axes of order-3 rotation symmetry pass (cfr. below). These points are highlighted with red dots in Figure 1 of the main article. Using (4) we write

$$\left(\frac{X}{3}\,\frac{X}{3}\right) = \frac{X}{3}\mathbf{A}_1 + \frac{X}{3}\mathbf{A}_2 \tag{9}$$

$$= \frac{X}{3}\left(k - s\right)\mathbf{a}_1 + \frac{X}{3}\left(k + 2s\right)\mathbf{a}_2 \tag{10}$$

where the integer $X = 1$ or $2$ selects the supercell high symmetry point. Let us introduce now the integer parameter $\alpha$ defined as

$$k - s = 3t + \alpha$$

with $t \in \mathbb{Z}$, so only $-1$, $0$ and $1$ are meaningful values of $\alpha$. Using it in equation (10), we get

$$\left(\frac{X}{3}\,\frac{X}{3}\right) = \frac{X}{3}\left(3t + \alpha\right)\mathbf{a}_1 + \frac{X}{3}\left(3t + 3s + \alpha\right)\mathbf{a}_2 \tag{11}$$

$$= \underbrace{Xt\,\mathbf{a}_1 + X(t+s)\mathbf{a}_2}_{=\mathbf{R}} + \frac{X\alpha}{3}\left(\mathbf{a}_1 + \mathbf{a}_2\right) \tag{12}$$

where $\mathbf{R}$ is a honeycomb lattice vector. Therefore, if $\alpha = -1$ and $X = 1$, the site located in $(\frac{1}{3}\,\frac{1}{3})$ of the *supercell* will coincide with the site located at $(-\frac{1}{3}\,-\frac{1}{3}) = (\frac{2}{3}\,\frac{2}{3})$ of the *primitive cell* of the honeycomb lattice, and vice-versa if $X = 2$. But if $\alpha = +1$, the site in $(\frac{1}{3}\,\frac{1}{3})$ will coincide with the site in $(\frac{1}{3}\,\frac{1}{3})$ of the primitive cell, and the same for $X = 2$. Actually, we demonstrate below in the Supplementary Materials that the case $\alpha = 0$ is irrelevant.

Lastly, it is easy to demonstrate that if a given supercell $(p, q)$ has a $\alpha = +1$ parameter, then the supercells $(q, p)$ and $(-q, p + q)$ have a $\alpha = -1$ parameter (and inversely).

### B: Stacking geometries

As we mentioned in the main article, our construction of the moiré geometries requires two integers $\{p, q\}$ and follows the rules: (i) the lower layer is always defined by the $(q, p)_{\mathrm{B}}$ supercell (origin at boron) and (ii) the upper layer is either defined by the $(p, q)_{\mathrm{X}}$ cell or the $(-q, p + q)_{\mathrm{X}}$ cell, where X labels the origin of the supercell (B = boron, N = nitrogen, H = hexagon center). As shown in the previous section, the $(p, q)$-on-$(q, p)$ constructions will always be made of supercells with opposite $\alpha$ parameters, whereas the $(-q, p+q)$-on-$(q, p)$ constructions will always result from supercells with the same $\alpha$. The Table IV lists the kind of sublattice (boron, nitrogen atom, or hexagon center) that occurs at the high symmetry points for both values of $\alpha$ of the lower layer $(q, p)_{\mathrm{B}}$.

| supercell | $\alpha$ | $(0\,0)$ | $(\frac{1}{3}\,\frac{1}{3})$ | $(\frac{2}{3}\,\frac{2}{3})$ | $\alpha$ | $(0\,0)$ | $(\frac{1}{3}\,\frac{1}{3})$ | $(\frac{2}{3}\,\frac{2}{3})$ | name of the bilayer obtained |
|---|---|---|---|---|---|---|---|---|---|
| $(q,p)_{\text{B}}$ | -1 | B | H | N | +1 | B | N | H | by stacking on the $(q,p)_{\text{B}}$ |
| $(p,q)_{\text{B}}$ |  | B | N | H |  | B | H | N | BB$(q,p)$ |
| $(p,q)_{\text{N}}$ | +1 | N | H | B | -1 | N | B | H | BNNB$(q,p)$ |
| $(p,q)_{\text{H}}$ |  | H | B | N |  | H | N | B | NN$(q,p)$ |
| $(-q,p+q)_{\text{B}}$ |  | B | H | N |  | B | N | H | BBNN$(q,p)$ |
| $(-q,p+q)_{\text{N}}$ | -1 | N | B | H | +1 | N | H | B | BN$(q,p)$ |
| $(-q,p+q)_{\text{H}}$ |  | H | N | B |  | H | B | N | BN$(q,p)$ |

TABLE IV. Determination of the kind of the sublattices located at the high symmetry points used in our construction of bilayers for a generic $\{p,q\}$ pair, and the name of the resulting bilayer.

For any choice of $p$ and $q$, the six possible stackings are:

1. The $(p,q)_{\text{B}}$-on-$(q,p)_{\text{B}}$ is a single coincidence structure, with B on B at the origin, N on hexagon at one of the two high-symmetry points and a hexagon on N at the other one. There is no hexagon-on-hexagon vertical alignment for the single coincidence structures. We call this structure the BB$(q,p)$ bilayer.

2. The $(p,q)_{\text{N}}$-on-$(q,p)_{\text{B}}$ is a double coincidence structure, with N on B at the origin, B on N at one of the two high-symmetry points and an hexagon-on-hexagon at the other one. We call it the BNNB$(q,p)$ bilayer.

3. The $(p,q)_{\text{H}}$-on-$(q,p)_{\text{B}}$ is again a single coincidence structure, with a hexagon on B at the origin, B on hexagon at one of the two high-symmetry points and an N on N at the other one. We call it the NN$(q,p)$ bilayer.

4. The $(-q,p+q)_{\text{B}}$-on-$(q,p)_{\text{B}}$ is another double coincidence structure, with B on B at the origin, N on N at one of the two high-symmetry points and an hexagon-on-hexagon at the other one. We call it the BBNN$(q,p)$ bilayer.

5. The $(-q,p+q)_{\text{N}}$-on-$(q,p)_{\text{B}}$ is a single coincidence structure, with N on B at the origin, N-on-hexagon at one of the two high-symmetry points and an B-on-hexagon at the other one. We call it the BN$(q,p)$ bilayer.

6. The $(-q,p+q)_{\text{H}}$-on-$(q,p)_{\text{B}}$ is a single coincidence structure, with a hexagon on B at the origin, N on hexagon at one of the two high-symmetry points and an B on N at the other one. It is the same geometry than the BN$(q,p)$ above.

Finally, since the stacking 6 leads actually to the same structure as stacking 5, for each $\{p,q\}$ pair of integer we construct five and only five different structures that preserve the atom-on-atom vertical alignments.

### C: Moiré stacking angles

The easiest way to derive the twist angle between two bilayers is by representing the vectors of the honeycomb lattice with discrete complex numbers. Here, we adopt the notation [41, 43] $\mathcal{Z}(m,n) = mz_1 + nz_2$ with $z_1 = 1$ and $z_2 = \frac{1}{2} + \frac{\sqrt{3}}{2}i$. The angles are just the arguments calculated like

$$\exp(i\theta) = \frac{\mathcal{Z}(q,p)}{\mathcal{Z}(p,q)} \tag{13}$$

$$\exp(i\theta') = \frac{\mathcal{Z}(-q,p+q)}{\mathcal{Z}(q,p)} \tag{14}$$

and depend only on the $\{p,q\}$ pair of integers. This leads to

$$\tan\theta_{\{p,q\}} = \sqrt{3}\,\frac{p^2 - q^2}{p^2 + q^2 + 4pq} \tag{15}$$

$$\tan\theta'_{\{p,q\}} = \sqrt{3}\,\frac{q^2 + 2pq}{2p^2 - q^2 + 2pq} \tag{16}$$

which are given in the main article. Since the $p$ and $q$ indices can take any integer value, the angles are always defined modulo 60°. The constructed supercells and the resulting angles $\theta$ and $\theta'$ are drawn in Figure 6.a.

So far, the vectors defined by (2) have been developed on the $\{\mathbf{a}_1, \mathbf{a}_2\}$ honeycomb lattice basis, but we could have chosen either to develop them on the $\{\mathbf{a}_2 - \mathbf{a}_1, -\mathbf{a}_1\}$ basis and then work with the $\{-p-q,p\}$ pair, or on the $\{-\mathbf{a}_2, \mathbf{a}_1 - \mathbf{a}_2\}$ basis, and work with the $\{q, -p-q\}$ pair. So, definitions (15) and (16) are not unique and the angles could have also been defined as

$$\tan\theta_{\{-p-q,p\}} = \sqrt{3}\,\frac{q^2 + 2pq}{-2p^2 + q^2 - 2pq} \tag{17}$$

$$\tan\theta'_{\{-p-q,p\}} = \sqrt{3}\,\frac{-p^2 - 2pq}{-p^2 + 2q^2 + 2pq} \tag{18}$$

or

$$\tan\theta_{\{q,-p-q\}} = \sqrt{3}\,\frac{p^2 + 2pq}{-p^2 + 2q^2 + 2pq} \tag{19}$$

$$\tan\theta'_{\{q,-p-q\}} = \sqrt{3}\,\frac{-p^2 + q^2}{p^2 + q^2 + 4pq} \tag{20}$$

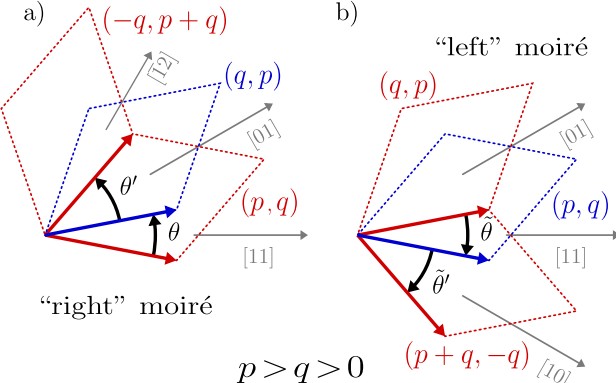

FIG. 6. a) The angles $\theta$ and $\theta'$, based on the $(q,p)$ geometries (that are used in the main article). b) The angles $\tilde{\theta}$ and $\tilde{\theta}'$ corresponding to the mirror images of the previous ones. They are based on the $(p,q)$ geometries.

which are also valid formulations. It is trivial to show that for any $\theta$ of equations (15), (17), or (19) and for any $\theta'$ of equations (16), (18), or (20), the following equality

$$\theta' = -\theta + \frac{n\pi}{3}$$

holds for an integer $n \in \mathbb{Z}$. In order to avoid confusion and give a non ambiguous definitions of our moiré structures, we decide arbitrarily to adopt definitions (15) and (16), and to impose

$$p > q > 0.$$

In this situation, the vectors $p\mathbf{a}_1 + q\mathbf{a}_2$ and $q\mathbf{a}_1 + p\mathbf{a}_2$ lie in the $\{\mathbf{a}_1, \mathbf{a}_2\}$ angular sector, and the vector $-q\mathbf{a}_1 + (p+q)\mathbf{a}_2$ lie in the $\{\mathbf{a}_2, \mathbf{a}_2 - \mathbf{a}_1\}$ angular sector. As a consequence

$$\theta, \theta' \in \left]0, \frac{\pi}{3}\right[ \quad \text{and} \quad \theta + \theta' = \frac{\pi}{3}$$

implying that $\mathrm{BB}(q,p)$, $\mathrm{BN}(q,p)$ et $\mathrm{NN}(q,p)$ have an angle $+\theta > 0$ and $\mathrm{BBNN}(q,p)$, $\mathrm{BNNB}(q,p)$ have an angle $-\theta' < 0$. These five stackings are chiral structures, that we decide to name "right" moiré bilayers.

To construct the enantiomers of the "right" moiré bilayers above, we have to transform the vectors $\mathbf{A}_1$ defining the hexagonal supercells (2). They are mirrored respect the $[1\,1]$ crystallographic direction of the primitive honeycomb lattice cell, as shown in the Figure 6.b. The lower layer of a "left" moiré is now carried by the supercell $\mathrm{M}^{(p,q)}$ and the upper layer is developed either on the $\mathrm{M}^{(q,p)}$ or the $\mathrm{M}^{(p+q,-q)}$ one, still within the constraint $p > q > 0$. The corresponding twist angles are now

$$\exp(i\tilde{\theta}) = \frac{\mathcal{Z}(p,q)}{\mathcal{Z}(q,p)} \tag{21}$$

$$\exp(i\tilde{\theta}') = \frac{\mathcal{Z}(p+q,-q)}{\mathcal{Z}(p,q)} \tag{22}$$

leading to $\tilde{\theta} = -\theta$ and $\tilde{\theta}' = -\theta'$ then

$$\tilde{\theta}, \tilde{\theta}' \in \left]-\frac{\pi}{3}, 0\right[ \quad \text{and} \quad \tilde{\theta} + \tilde{\theta}' = -\frac{\pi}{3}.$$

As a result, the "left" $\mathrm{BB}(p,q)$, $\mathrm{BN}(p,q)$ and $\mathrm{NN}(p,q)$ have an angle $-\theta < 0$, and the "left" $\mathrm{BBNN}(p,q)$ and $\mathrm{BNNB}(p,q)$ have an angle $+\theta' > 0$.

In absence of any magnetic field, the "right" and "left" corresponding stackings exhibit exactly the same electronic properties. That is why we restricted our study to the "right" ones.

## D: Redundancy of the case $(p - q = 3t)$

The case $\alpha = 0$ corresponds to moiré $(p,q)$ supercells where $p - q = 3t$ and $t$ is an integer. So

$$\begin{bmatrix} q+3t & q \\ -q & 2q+3t \end{bmatrix} = (q+3t, q) \text{ supercell.} \tag{23}$$

As we sketched in figure 7, starting from the vectors $\mathbf{A}_1$ and $\mathbf{A}_2$, we can define new shorter vectors

$$\mathbf{v}_1 = \frac{2}{3}\mathbf{A}_1 - \frac{1}{3}\mathbf{A}_2 = (q+2t)\,\mathbf{a}_1 - t\,\mathbf{a}_2 \tag{24}$$

$$\mathbf{v}_2 = \frac{1}{3}\mathbf{A}_1 + \frac{1}{3}\mathbf{A}_2 = t\,\mathbf{a}_1 + (q+t)\,\mathbf{a}_2 \tag{25}$$

$$\mathbf{v}_3 = -\frac{1}{3}\mathbf{A}_1 + \frac{2}{3}\mathbf{A}_2 = (-q-t)\,\mathbf{a}_1 + (q+2t)\,\mathbf{a}_2 \tag{26}$$

and since $q$ and $t$ are integers, the vectors $\mathbf{v}_i$ are honeycomb bravais lattice vectors. In this situation, the supercell defined by the indices of the vector $\mathbf{v}_3$ (for example) is

$$\begin{bmatrix} -q-t & q+2t \\ -q-2t & t \end{bmatrix} = (-q-t, q+2t) \text{ supercell} \tag{27}$$

which is also an asymetric hexagonal supercell, three times smaller than the original $(q+3t, q)$ one.

Moreover, the twist angles (15) calculated with $p$ and $q$ indices (when $p = q + 3t$) are

$$\tan\theta_{\{q+3t,q\}} = \sqrt{3}\,\frac{3t^2 + 2qt}{2q^2 + 3t^2 + 6qt}$$

$$\tan\theta'_{\{q+3t,q\}} = \sqrt{3}\,\frac{q^2 + 2qt}{q^2 + 6t^2 + 6qt}$$

and it is staightforward to verify than these two tangents are exactly the same if we calculate them with the $-q-t$ and $q+2t$ indices.

To summarize, (i) the $\{q+3t, q\}$ set leads to non primitive moiré supercells, and (ii) it is always possible to use the $\{-q-t, q+2t\}$ pair which gives the same twist angles but in three times smaller supercells. As an illustration of it, in Figure 8 we have drawn the example of the construction of the $(-1,5)$-on-$(1,4)$ moiré and its reduction to the $(1,2)$-on-$(2,1)$ "left" moiré bilayer.

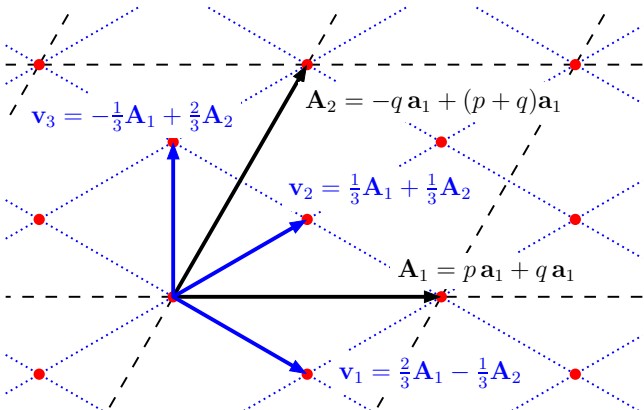

FIG. 7. The upper layer asymmetric supercell $(p, q)$ with $p = q + 3t$. It is always possible to construct a smaller supercell since $\mathbf{v}_1$, $\mathbf{v}_2$ and $\mathbf{v}_3$ are vectors of the honeycomb lattice. In other words, the twisted bilayer geometries constructed from the $(q, q + 3t)$ supercell are not primitive cells of the moiré.

## E: Layer groups of moiré structures

In Figure 9, we report graphical representations of the symmetries of the layer group used in this Appendix. The layer group of a graphene monolayer asymmetric supercell is the $p6/m$, neglecting translations occurring inside the defined cell. For a boron nitride (or a transition metal dichalcogenide) supercell, the layer group is $p\bar{6}$ [52]. Both groups contain order-3 or order-6 rotations axis along $z$, located at the high symmetry points of the cell: $(0\,0)$, $\left(\frac{1}{3}\,\frac{1}{3}\right)$ and $\left(\frac{2}{3}\,\frac{2}{3}\right)$. When stacking two supercells like described in the previous sections, these axes are coincident, and the rotations are always preserved. Thus the 2D crystal systems remain hexagonal.

By looking at Table IV and by replacing all occurrences of B and N by C, it is easy to derive all the stackings of graphene bilayers, however the result is highly redundant. Actually, by taking the origin of all the supercells only on the site corresponding to B atoms in hBN, it is possible to sort out identical geometries from the beginning. In this case, the $(-q, p + q)$-on-$(q, p)$ structure geometry always shows one "hexagon-on-hexagon" vertical alignment with an order-6 rotation axis, and two atom-on-atom vertical alignments with order-3 rotation axis (*double* sublattice coincidence). The resulting layer group is the hexagonal $p622$, that also contains many in-plane order-2 rotations, oriented along $[1\,0]$ and $[1\,1]$ crystallographic directions as well as many $2_1$ screw axes. Note that to comply with the definitions of layer group as defined in Figure 9, the supercell must have the "hexagon-on-hexagon" axis is located at the origin. This means that supercells constructed as we have done in our work must be translated accordingly. Differently, the case of $(p, q)$-on-$(q, p)$ structure exhibits two "hexagon-on-atom" alignments and one "atom-on-atom" alignment (*single*

sublattice coincidence) in the points where order-3 rotation axes pass. If the structure is constructed like proposed above in this Supplementary Material, this "atom-on-atom" coincidence is correctly located at the origin. It is worth noticing that there are in-plane order-2 rotations axes, oriented along the $[1\,0]$ crystallographic directions, passing through the origin. The symmetry group is $p321$ for this case.

Let now analyze the symmetry of the hBN moiré bilayers. As explained in the previous sections, the three stackings $BB(q, p)$, $NN(q, p)$, and $BN(q, p)$ correspond geometrically to the graphene bilayer with *single* sublattice coincidence. Note that, as previously, the NN stacking must be translated in such a way that the "atom-on-atom" vertical coincidence is placed at the origin, while this is not needed for the other two stackings that result constructed consistently. The BB and the NN stacking geometries keep the in-plane order-2 rotations axes along $[1\,0]$. Therefore their layer group is also the $p321$. However, in the BN stacking case, the coincident atoms are now chemically different and the order-2 rotations are lost. The group is the simplest hexagonal $p3$.

The last two hBN moiré stackings are the $BBNN(q, p)$ and the $BNNB(q, p)$ which correspond geometrically to the graphene *double* sublattice coincidence moiré. Again, we translate the structures to locate the "hexagon-on-hexagon" vertical axis at the origin. A careful observation of the $BNNB(q, p)$ moiré geometry allows us to notice that the in-plane order-2 rotation axes along $[1\,0]$ and passing through the origin are conserved. The layer group of the BNNB moiré stacking is then again the $p321$. Differently, in the $BBNN(q, p)$ structure, the in-plane order-2 rotation axes that are preserved are oriented along the $[1\,1]$ crystallographic directions. The layer group of symmetry of BBNN stacking is then the $p312$.

In this work, we have built structures paying attention to preserve the vertical atomic coincidence, and consequently the order-3 rotation axes. However, we can ask ourselves what happens if we stack a $(p, q)$ or a $(-q, p+q)$ supercell on a $(q, p)$ cell with a totally random translation between the layers. In this scenario, all the point symmetry operations are lost, and only the translations are preserved by construction. This implies that, although the supercell vectors have the same length and span an angle of 60°, the crystal system is no longer hexagonal. It is *oblique* and the layer group is the simplest $p1$. In the reciprocal plane, only the $+\mathbf{k}/-\mathbf{k}$ symmetry is conserved, and consequently the high-symmetry points K are no longer equivalent.

## F: Computational details

Calculations have been done with the free simulation packages Quantum ESPRESSO [53, 54] (band structure

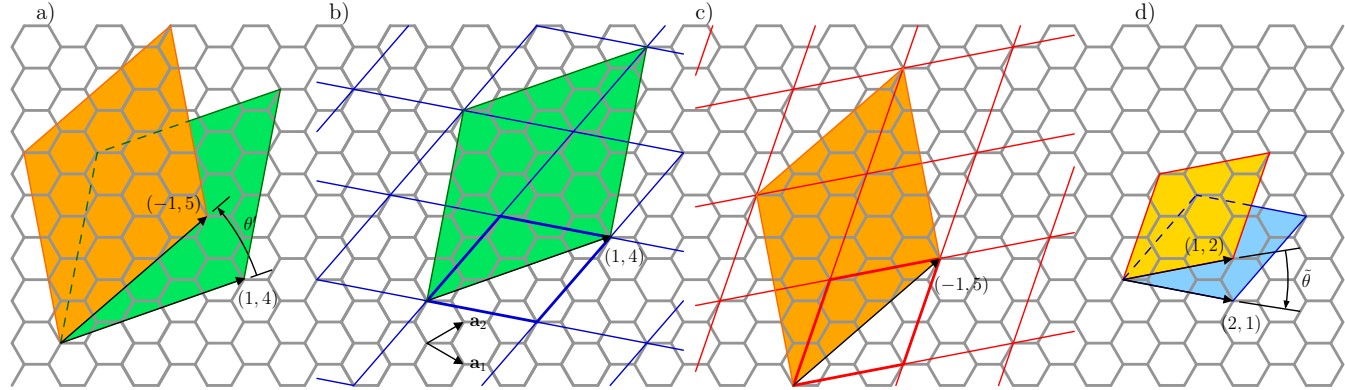

FIG. 8. a) Construction of the moiré bilayer based on the $(1,4)$ supercell for the lower layer and the $(-1,5)$ for the upper layer. b) The lower supercell can be tessellated by the $(2,1)$ smaller supercell. c) The upper one is also a tessellation of the $(1,2)$ supercell. d) The angle of the "left" small moiré is the same as that of the large non-primitive moiré $\tilde{\theta}_{\{2,1\}} = -\theta'_{\{1,4\}}$.

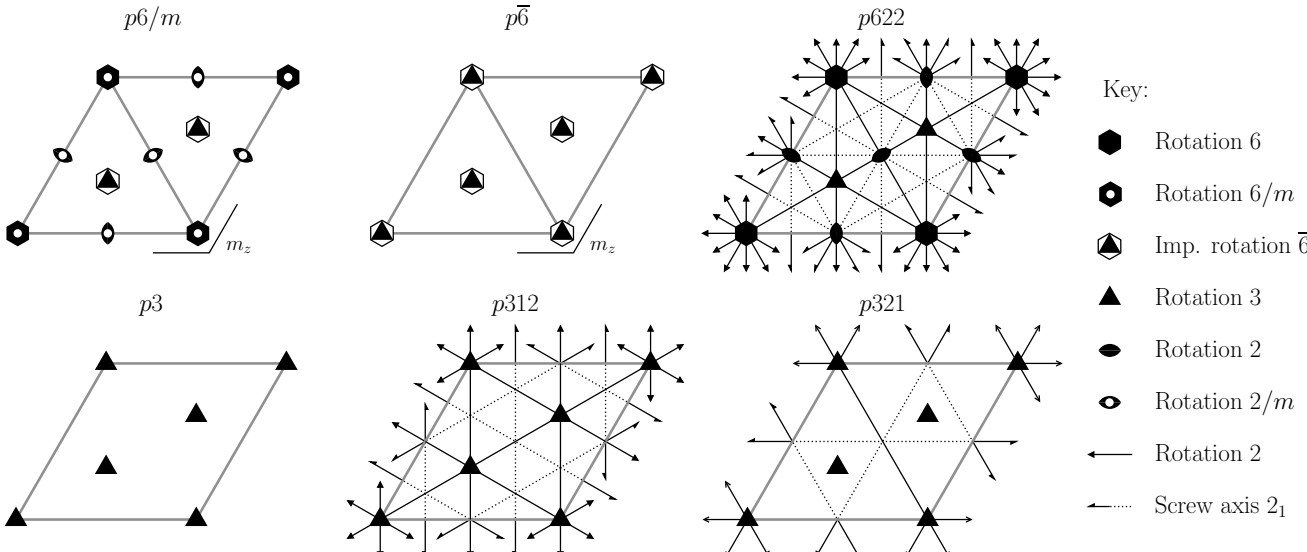

FIG. 9. The graphene and hBN moiré bilayers belong to one of these layer groups (adapted from [52]). The trivial group $p1$ is not shown.

of twisted bilayers) and ABINIT [55, 56] (stability of twisted and untwisted bilayers).

In both cases norm-conserving pseudopotentials have been used. We checked that switching from one software to the other was not introducing major errors in the main characteristics discussed in the paper. In both groups of calculations, the cutoff energy was 30 Ha and we sampled the Brillouin zone with a Monkorst-Pack grid of $5 \times 5 \times 1$ k-points in all supercells ($9 \times 9 \times 1$ in the untwisted cases). The equilibrium interlayer distance has been fixed at 3.22 Å in all bilayers as detailed below. The in-plane cell parameter was $a = 2.23$ Å and no in-plane relaxation has been done. A cell height $L = 15$ Å has been used in all calculations unless specified differently. This value has been fixed by paying attention to the alignment of the $\sigma$ and $\pi$ conduction bands. In fact, as already pointed out

by several authors [15, 44–49] the bottom conduction in $\Gamma$ is composed of nearly-free-electron (NFE) states that extend for several Ångströms above the layer and thus converge very slowly with the amount of vacuum (see the dedicated section of the Supplemental Material).

To fix the interlayer distance, we calculated the total energy per unit formula $E(h)$ at different input values of the interlayer distance $h$. Results are reported in Figure 10. We took the BB(1,2) and the BB(2,3) bilayers as reference structures. For these bilayers, we sampled $h$ on a fine grid. Both bilayers have the energy minimum at $h = 3.22$ Å, with a negligible energy difference ($\sim 0.1$ meV per formula unit). Then we computed $E(h)$ for the BN(1,2), NN(1,2), BNNB(1,2) and BBNN(1,2) bilayers on a coarser grid and found that the points fell basically on top of the BB(1,2) curve. Following this analysis, we

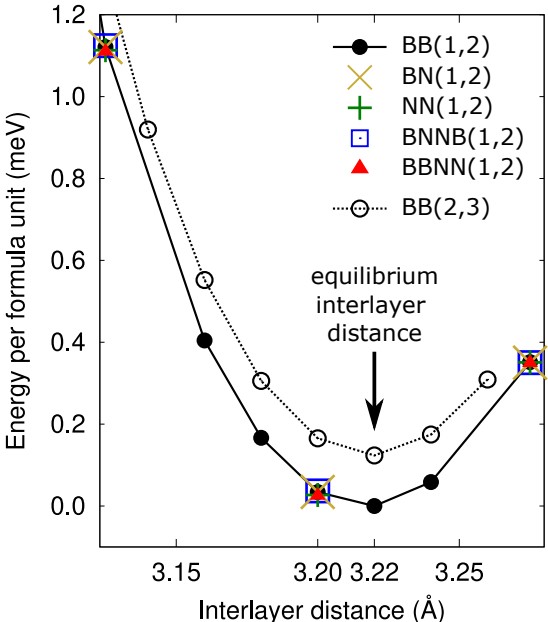

FIG. 10. Total energy calculation of the five stackings in the (1,2) supercell as a function of the interlayer distance $h$. The BB(1,2) is the full black line with black bullets and the BB(2,3) is the dotted line with empty circles. The other (1,2) stackings are superimposed to the BB(1,2) curve almost exactly and are reported with different colors and symbols.

deduced that we can safely fix the equilibrium distance at $h = 3.22$ Å irrespective of the stacking or the twist angle. We note however that this value may be inaccurate for very small twist angles that are not investigated in this work.

### G: Untwisted bilayers

It is possible to extend the nomenclature we introduced in the main text to untwisted bilayers. In this case, only the stacking label is meaningful, the $(q, p)$ pair being trivially 1 and 0. In Figure 11 we report an image

| System | $h$ | $E_{BN}$ | $E_{ind}$ | $E_{dir}$ |
|---|---|---|---|---|
| BBNN(0,1) | 3.425 | 8.7 | 3.957 | 4.037 |
| NN(0,1) | 3.375 | 6.8 | 4.345 | 4.037 |
| BB(2,3) | 3.220 | 0.1 | 4.217 | 4.251 |
| **BB(1,2)** | **3.220** | **0** | **4.318** | **4.394** |
| BB(0,1) | 3.150 | -8.3 | 3.950 | 4.436 |
| BNNB(0,1) | 3.125 | -11.1 | 4.649 | 4.398 |
| BN(0,1) | 3.100 | -12.8 | 4.463 | 4.438 |

TABLE V. Equilibrium interlayer distance $h$ (Å), total energy per formula unit $E_{BN}$ with respect to the BB(1,2) bilayer (in meV) , smallest indirect gap $E_{ind}$ (eV) and energy of the smallest direct transition $E_{dir}$ (eV) (direct gap).

of the structure of the five untwisted stackings and their stability curve $E(h)$ together with that of the BB(1,2) bilayer. We observe that the three most stable untwisted structures, i.e. the BN(0,1), the BNNB(0,1) and the BB(0,1) have a smaller equilibrium distance, whereas for the two most unstable, the NN(0,1) and the BBNN(0,1), the equilibrium $h$ is larger, so that the twisted bilayers fall somewhat between the two groups. This makes sense if one reckons that inside the same twisted bilayer one can find domains with a local stacking intermediate to the five untwisted ones.

In experiments it is observed that, far from certain angles, it is pretty easy to move or twist a BN flake on top of another, and this is consistent with the negligible energy differences we calculated between different stackings at fixed angle and between the two reference calculations with the same stacking sequence. However when the twist angle gets close to some specific values, the flake gets stuck and no further twist is possible. In fact, the large energy differences with the untwisted configurations (order of 10 meV per unit formula) suggest that when approaching small twist angles the bilayer falls into one of the energetically more favorable configurations, possibly undergoing large in-plane deformation to maximize the size of the untwisted domains. [30, 50, 51, 57].

The equilibrium distances, the total energy per BN pair with respect to the BB(1,2) bilayer and the values of the DFT direct (at K) and indirect band gaps (between valleys close to K and the point M) are reported in Table V.

### H: Nearly-free-electron states

As already pointed out by Blase and coworkers in the case of bulk hBN [15], the conduction states at Γ converge very slowly with the amount of vacuum because they correspond to some unoccupied N-centered nearly-free-electron (NFE) state extending for several Ångströms above the BN layer [15, 44–49]. These NFE states have a neat $3s$ orbital component, as shown in the fat-band plot reported in Figure 12.

Their alignment with respect to the π bands is a delicate issue on the purpose of this article because the energy difference between the bottom of the unoccupied σ band and the bottom of the unoccupied π band are very close in energy and they may compete in determining the indirect nature of the gap. Therefore, it is worth paying much attention to their convergence. To this aim, we made a series of two test calculations in a BN(1,2) bilayer. First we tested the evolution of these states as a function of the height of the simulation cell at fixed interlayer distance (the three panels of Figure 13a). This test shows that by reducing the cell height, the NFE states are pushed toward higher energies because of fictitious cell-to-cell interactions. Replicas of the system must be

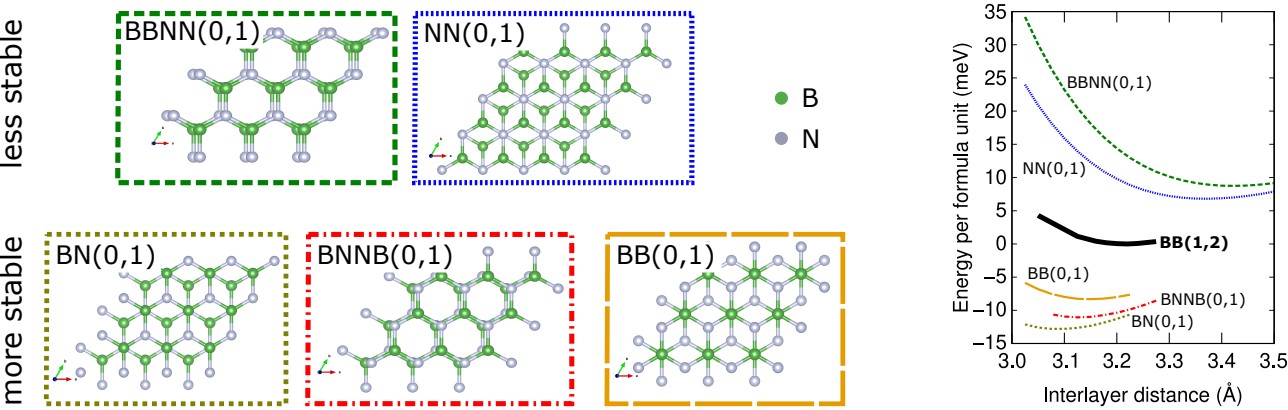

FIG. 11. The five hexagonal stackings in untwisted bilayers and their stability curves with respect to the BB(1,2) twisted bilayer.

separated of around $L \sim 20$ Å for the band dispersion and alignment to be converged. Note that we decided on purpose to carry out our simulations with a slightly lower value (15 Å) because the fact of pushing the NFE states to higher energies is not detrimental to our investigation and allows us to reduce the computational workload.

Then we tested the evolution of the NFE states as a function of the interlayer distance leaving a constant amount of vacuum $(L - h)$ of 40 Å, which is largely enough to prevent cell-to-cell interactions. In the panels of Figure 13.b, we report three calculations of the BN(1,2) bilayer with a varying interlayer distance (20, 10 and 7.5 Å respectively in panels b1, b2 and b3). In the b1 panel, we also plot in black the conduction band of the isolated monolayer in the (1,2) supercell and we verify that it coincides with the $h = 20$ Å bilayer calculation. This test demonstrates that moving two layers closer to each other induces a bonding/antibonding splitting of the NFE states which increases as the layers get closer.

Since there is no difference between the interlayer dis-

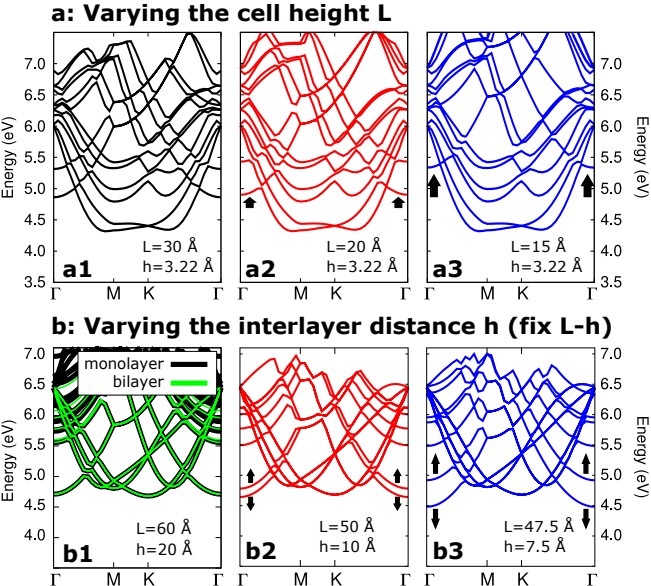

FIG. 13. The evolution of the NFE states as a function of the simulation parameters in the BN(1,2) bilayer. **a**: evolution as a function of the cell height $L$ at fixed interlayer dsitance ($h = 3.22$ Å). $L = 30$, 20 and 15 Å in panels **a1**, **a2** and **a3** respectively. **b**: evolution as a function of the interlayer dsitance $h$ at fixed vacuum ($L - h = 40$ Å). $h = 20$, 10 and 7.5 Å in panels **b1**, **b2** and **b3** respectively. In panel **b1**, the band structure of the BN(1,2) bilayer (flashy green) is compared with that of the isolated monolayer (black).

tance separating two layers inside the cell and the space separating replicas of the simulated system, one should pay attention that these two effects (pushing to higher energies and band splitting) happen at the same time.

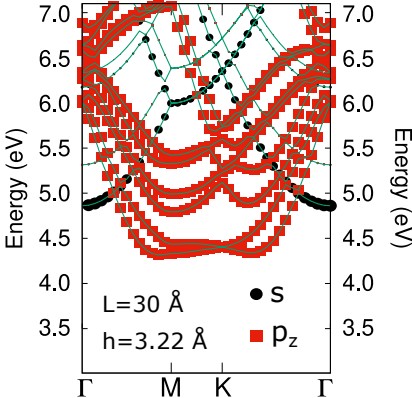

FIG. 12. Orbital momentum component of the conduction bands of the BN(1,2) bilayer (fat bands).

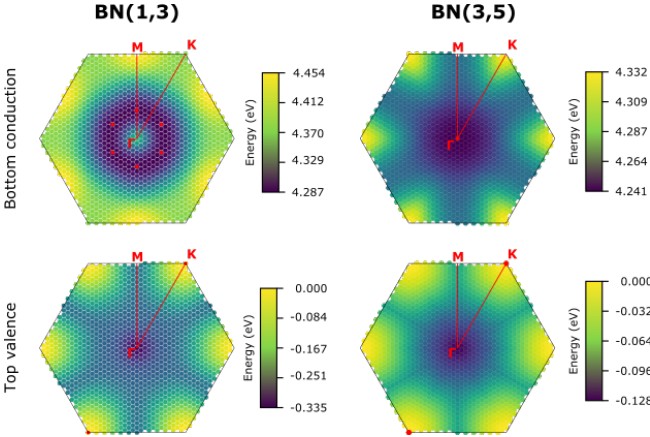

FIG. 14. Energy surface of the lowest empty band (top panels) and the highest occupied band (bottom panels) of the BN(1,3) and the BN(3,5) bilayers from left to right. The top valence and the bottom conduction states are highlighted with red hexagons.

### I: Band gap of the $\delta = 2$ family

In the main text we give the values of the gapwidth of the five stackings of the (1,3) and (3,5) supercells. The values have been extracted from the corresponding band plots, so they refer to gapwidths calculated along specific high symmetry paths in the Brillouin zone. In this section we report a more complete mapping of the band structure of the top valence and bottom conduction of the BN stacking, chosen as representative of the bilayers. In Figure 14 we report the energy surface of the highest occupied states and the lowest unoccupied states in the BN(1,3) and BN(3,5) bilayers. With this analysis we demonstrate that the values reported in the main text are meaningful because the bottom of the conduction and the top of the valence fall indeed on the high symmetry lines.

For this analysis we acknowledge F. Paleari who kindly provided us with a dedicated analysis post-processing tool.

### J: Band structure of the other stackings

Here below we report the band plots missing in the main text corresponding to stackings BBNN, BB and BNNB from top to bottom.

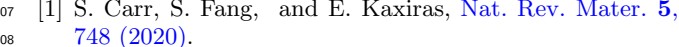

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
