# Peer review of "Structural classification of boron nitride twisted bilayers and ab initio investigation of their stacking-dependent electronic structure"

_SciPost Physics_

## Round 1 · Referee Report · Anonymous (Referee 1) · 2022-10-7

Strengths

  • interesting field of research
  • very neat geometrical analysis
  • proposition of a nomenclature to clarify the way we compare systems
  • ab initio analysis of the related properties

Weaknesses

  • the properties of twisted hBN bilayers at magic angles preserving hexagonal symmetry do not seem very "exciting" (no evolution from direct-to-indirect gaps)

Report

The authors present a geometrical analysis of bilayer hexagonal boron-nitride, focusing on magic angles preserving the hexagonal symmetry. The geometrical analysis and related nomenclature are very neat and the paper well written. Further, the electronic properties associated with these specific systems is explored at the DFT level, showing that the gap remains indirect for these angles preserving hexagonal symmetry.

As such, the paper certainly deserve publications with minor comments.

The paper is motivated by a nice introduction where the impressive case of twisted graphene is quickly reviewed, together with the modification of optical properties of twisted bilayer semiconducting 2D systems such as dichalcogenides. Similarly, a modification of the luminescence intensity (Ref[32]) of twisted hBN bilayers has been observed experimentally, motivating the present study (Ref.[32]). This is may be a limitation of the paper that the impact of twisting hBN bilayers is somehow much less important as compared to graphene. Further limiting the twist angle to values preserving the hexagonal symmetry further restricts the potential properties that can be obtained. As observed by the authors, smaller twist angles may allow tuning the direct/indirect nature of the gap, with important consequences on luminescent properties, but preserving the hexagonal symmetry leads to indirect band gap. If the authors can strengthen their motivations, this would certainly add value to the paper.

The authors observe that their DFT calculations contradict quite significantly previous density-functional tight-binding results. Since DFTB allows exploring much larger unit cells, providing more flexibility in tuning the twist angle, this is quite an unfortunate situation. Would the author have some intuition on what may have gone wrong (or what is missing) in the DFTB parametrization ? The referee understands that it is difficult to speculate on the study by another group.

Concerning the evolution of the band structure, several effects may be invoked : a change in hybridization, a change in the ionic potential (a purely electrostatic effect) and a change in interlayer screening that may affect differently states with different localization. The last effect is not included properly in DFT calculations but some of the authors are expert in this field. It would be quite informative to read about the origin of the differential in evolution between direct and indirect gaps. The referee doesn’t remember reading such an analysis concerning the direct (monolayer) to indirect gap evolution of h-BN in its simple AA’ stacking (e.g. Refs. [31,38]) so that such an analysis may be difficult to handle. Purely ionic effects may be possibly obtained by fixed classical charges on one layer.

The referee could not find in the main manuscript which functional is used. This is really unfortunate as it gives to non-specialist readers the feeling that DFT is a unique method, while the authors know very well that significant differences may stem from changing the functional. As a matter of fact, the referee is not fully convinced that changing the functional (e.g. going to an hybrid, global or range-separated) may only change the magnitude of the gap without changing the dispersion (as suggested in their discussion about GW calculations), and possibly the direct/indirect nature of the gap. The authors should really say which functional is used and discuss about changes in dispersion/bandwidth upon going away from (semi)local functionals.

Since SciPost somehow invites to question the ways we publish papers, the referee was a bit curious to understand the rational in selecting one theory paper out of very many concerning defects in hBN. Since this is not the central topic of the present paper, there is clearly no point making an exhaustive (impossible anyway) list a related theory papers and the referee has no solutions on how to bypass this problem of having to select a very arbitrary subset amongst many valuable published papers.
  • validity: high
  • significance: good
  • originality: high
  • clarity: high
  • formatting: excellent
  • grammar: excellent

Author:  Lorenzo Sponza  on 2022-11-02  [id 2972]

(in reply to Report 1 on 2022-10-07)
Category:
answer to question
reply to objection
correction

We acknowledge receipt of the report of Referee 1 and we thank the referee for his/her positive advice on our work.

The first among referee's criticism concerns the fact that we limited our investigation to hexagonal supercells. From the report, we understand that the referee finds (i) that this choice reduces the number of accessible twist angles and (ii) that is responsible of the indirect band gap.

(i) In fact, limiting the study to hexagonal lattices imposes restrictions to the stackings, not the twist angles. This is what allows us to split the nomenclature in a stacking label (BB, BN, BNNB, BBNN, NN) and two supercell indexes (p,q), which are ultimately related to the twist angle and are not restricted by the hexagonal geometry. If we restricted our study to relatively large angles is because of the size of the cell, not because of the hexagonal geometry. We will make this point clearer in the new version of the manuscript by adding the following sentences at line 127 of the new version:
“It is important to stress that these stacking sequences are not related to the moire periodicity and do not impose any constraint on the mutual orientation of the two layers. The orientation and the stacking are two independent degrees of freedom in the design of the bilayer structure.”

(ii) The choice of working with hexagonal geometries stems from the choice of investigating hexagonal Boron Nitride (hBN). In hBN, the indirect band gap comes from interlayer interactions (as highlighted in our previous study [38]) and is therefore a quite robust and general characteristics of multilayers based on hexagonal BN layers. Indeed, this feature has been reported in untwisted bilayers of different stackings [Mengle-Kioupakis_2019, Gilbert_2019], hBN multilayers [31], bulk of different stackings [38] and multilayers hosting different stacking sequences [Mengle-Kioupakis_2019].

On the other hand, experimental data [30, Moore_2021] as well as theoretical calculations with lattice relaxation [30, 50, 51] demonstrate that atoms rearrange to form domains with hexagonal symmetry. From there, the choice to focus on hexagonal stackings is clearly pertinent.

Later on, the referee asks about the discrepancy between our DFT results and those published with a DFT tight-binding method where the model is based on parameters to be fitted. It is indeed difficult to identify the origin of the difference, but in our experience [38] the indirect bandgap is essentially due to the interlayer interaction. It is possible that the parameters used by authors do not take correctly into account the B-B, B-N and N-N interlayer interactions.

We are not sure we understand referees's comment about the evolution of the band structure. The referee seems to ask us to go deeper in the explanation of the origin of the indirect gap in the bilayer, and on the reduction of the difference between direct and indirect band gap as a function of the twist angle. In [38] we examined the dispersion of the exciton in the monolayer and in three stackings of the bulk phase by means of GW+BSE calculations and an excitonic tight-binding model. We found that the indirect dispersion of the bulks is recovered by the tight-binding model only once the second-neighbour interlayer interaction is included in the model. We will make this statement clearer in the main text.
Concerning the difference between indirect and direct band gap, is a consequence of the flattening of the bands common to any twisted system.

The referee is definitely right about the absence of the details on the exchange correlation potential and we apologise for this forgetting which was not intentional. We used the PBE potential with Grimme-D2 scheme for van der Waals corrections. We will add this information in the corresponding Appendix.

The referee expresses some doubts on our statement that GW or other functionals (e.g. hybrids) will not change dramatically the dispersion. However, band dispersions available in literature [Berseneva_2013, Mengle-Kioupakis_2019, 16, 31, 37] show that quasiparticle corrections are very well described by a rigid shift of the conduction bands, which is often used as a justification for the use of the scissor operator. We will add appropriate references in the main text. Of course, these calculations have been published on monolayers or untwisted systems, but being the physics of the corrections mostly the same in twisted and untwisted bilayers. As a result, we don't expect any spectacular effect on the dispersion using GW or other functionals.

We will add to the main text the following references in the appropriate context (see page 4 of the revisited manuscript):
[Mengle-Kioupakis_2019] : K. A. Mengle and E. Kioupakis APL MAter. 7, 021106 (2019)
[Moore_2021] : S. L. Moore et al., Nature Comm. 12, 5741 (2021)
[Gilbert_2019] : S. Gilbert et al., 2D Materials 6, 021006 (2019)
[Berseneva_2013] : N. Berseneva et al., Phys. Rev. B 87, 035404 (2013)

Anonymous on 2022-11-16  [id 3033]

(in reply to Lorenzo Sponza on 2022-11-02 [id 2972])
Category:
answer to question

To answer number 1) we add also that, pushed by the comments of referee 3, we show that every periodic twisted bilayer can be described with our notation. We added this point as an appendix in the new version of the manuscript.

---

## Round 1 · Referee Report · Anonymous (Referee 2) · 2022-10-11

Strengths

1 - Precise relations for constructing BN moiré superstructures are defined.
2 – The authors demonstrate the existence of five and only five stacking configurations preserving the hexagonal symmetry.
3 – A physically informative nomenclature for an unambiguous identification of BN moiré supercells is proposed.
4 – The electronic structure of BN moiré supercells, obtained via DFT simulations, is discussed both as a function of the twist angle and the stacking sequence.
5 – A simple structural parameter has been introduced for classifying regularities in the electronic structure.

Weaknesses

1 - The geometry and symmetries of the moiré supercells are discussed directly in the case of the BN bilayer instead of having being presented in a more general framework. 2- The discussion on the electronic structure of the twisted bilayers is poorly linked with previous literature results on untwisted layers.

Report

In this work, the authors provide a detailed description of the structure of BN moiré supercells and how their electronic structure depends on both twist angle and stacking. In the first part of the paper, after introducing in a very pedagogical manner the conditions necessary to construct moiré BN supercells, the authors demonstrate that only five twisted bilayer stacking configurations preserves the hexagonal symmetry of the monolayer. These structures are then identified with a very concise and informative nomenclature.
In the second part of the paper the authors study, via DFT simulations, the electronic structure of BN bilayers as a function of stacking and twist angle. This systematic study made it possible to identify regularities in the electronic structure related to the relative position of the atoms in the two layers. Furthermore, the authors show that twisted bilayers always present an indirect gap whose width decreases as the twist angle decreases. This result is in contrast to what has previously been reported from lower level electronic structure calculations.

The article is well structured and clearly written. It is worthy of praise that the authors have provided extensive appendices in which all demonstrations, geometrical construction and computational details are discussed.
The work appears to be free of basic errors and therefore in my opinion no major corrections are required. However, I would like to propose a few recommendations that could help make the work addressable to a wider audience of potential readers.

1 - The discussion of the geometry and symmetries of moiré supercells provided in the first part of the paper can be applied to all types of hexagonal crystals with two inequivalent sites per unit cell. The authors have chosen to link this discussion directly to the BN case, but it could have been presented more generally by considering two arbitrary A and B sites. The nomenclature of high-symmetry configurations could then be easily transferred to other types of 2D materials. The first part of the article might then probably gain an audience in the very large community of researchers working on twisted transition metal dichalcogenides.

3 – In the paper it is shown how untwisted layers can be discussed within the framework of twisted layers. The existence of five stacking configurations respecting the hexagonal symmetry for the primitive BN bilayer cell has been reported several time in the literature. Here the authors show that this is a particular case in a more general framework. I think this point should be made explicit in the text and Figure 2 could be discussed also in relation to figure 11.

4 – In figure 2 the authors show how high symmetry configurations can transform by applying a translation of the layers. Intermediate configurations have a lower symmetry and in the case of untwisted layers they are energetically unstable. IS this behavior conserved in the case of twisted layers?

5 - The authors introduce the parameter $\delta=|p-q|$ to classify twisted bilayers in families with similar band structures. For all the untwisted configurations $\delta=1$ and $\theta=0$. Adding this case to figure 5 and 15 would permit to link these new results with the past literature. An extensive discussion of the electronic structure of the untwisted bilayer, presenting also GW results, is given in Mengle and Kioupakis APL Mater. 7 021106 (2019). This reference could be added in the text.

A last minor comment

1 - Page 1 line 82-83 “...we first define coincidence supercell [40]”. I suppose the authors wanted to refer here to a different article.
  • validity: high
  • significance: high
  • originality: high
  • clarity: high
  • formatting: excellent
  • grammar: good

Author:  Lorenzo Sponza  on 2022-11-02  [id 2975]

(in reply to Report 2 on 2022-10-11)

We acknowledge receipt of the report of Referee 2 and we thank the referee for his/her positive advice and suggestions.

  1. We thank the referee for the suggestion 1. It is actually not a big work to rename N sites as A (B being already B), but we should change all labels, figures and formulas. We frankly prefer to leave this work to the reader. However we will highlight better the generality of our geometrical analysis in the main text (see added sentences in page 2 and 3 of the new version of the manuscript).

  2. We thank the referee for this comment which is indeed very appropriate. We will make the modifications suggested by the referee in the new version of the paper.

  3. Indeed, we have not studied intermediate twisted configurations. Part of the reason is because it is known experimentally that massive reconstructions take place at small twist angles to maximise the size of more stable untwisted-like domains [30, 50, 51, Moore_2021]. On the other hand, we expect reconstructions to be less important at large twist angles.

  4. We thank the referee for this comment and the reference we overlooked. The referee is right in pointing out that the untwisted cases fall formally in the family delta=1. However, if the notation STACK(q,p) is appropriate for the atomic structure of all cases including the untwisted bilayers, their electronic structure seems to be unique in many respects. Some features actually differ from the twisted ones of the same family and we think that including a comparison may mislead the reader. For example, the conduction band of the BN(0,1) bilayer does not have the band crossing displayed by the twisted systems. We don’t have an ultimate explanation for this unicity, but it must be related to the specific symmetry properties of the untwisted bilayers. In fact, if in presence of a twist angle the BB, NN and BNNB stackings belong to the layer group #68 (p321), the BN to the #65 (p3) and the BBNN to the #67, at no twist they belong to layer groups #72, #69 and #78 respectively. In the main text, we added some sentences to explain why we don’t show the untwisted band structures, the basic symmetry consideration that we think participate to the explanation of their unicity, and we stressed that the notation STACK(q,p) can be generalized to untwisted bilayers to define their atomic structure but not to derive their electronic properties.

Concerning the last comment, we thank the referee for pointing out this error. In fact we wanted to cite another reference. We made a mistake in the bibtex file.

We will add to the main text the following references in the appropriate context: [Mengle-Kioupakis_2019] : K. A. Mengle and E. Kioupakis APL MAter. 7, 021106 (2019) [Moore_2021] : S. L. Moore et al., Nature Comm. 12, 5741 (2021) [Gilbert_2019] : S. Gilbert et al., 2D Materials 6, 021006 (2019) [Berseneva_2013] : N. Berseneva et al., Phys. Rev. B 87, 035404 (2013)

---

## Round 1 · Referee Report · Anonymous (Referee 3) · 2022-11-7

Report

The authors study the possible staking of BN twisted bilayers preserving hexagonal symmetry. They introduce clear and robust definitions and nomenclature, and study the effects on the band structure.
The manuscript is well written and structured. The figures are helpful and high quality. I would recommend publication, after the authors have addressed the following comments.

Requested changes

There are two issues that I believe many readers would wonder about. So I think they should be answered :

1) The absence of in-plane relaxation. It is well known to be important in BN-Graphene heterobilayer. The authors mention that relaxation could be important for small angles. Does that imply it is not for large ones? Have the authors tried to relax one test system? Are there other works in the literature that justify neglecting relaxation à priori?

2) The generality of the supercell definition. The choice of the matrix for generating the supercell comes off a bit abrupt. It is explicitely said to be arbitrary, but there must be some reasons behind it. I can see the supercell vectors are kept at the same norm, the angle between them seems conserved. Can the authors justify their choice a bit more? This is mostly to satisfy the curiosity of the reader. Another, more important point would be to justify the generality of this definition. Are all the possible supercells (preserving hexagonal symmetry) covered by this choice?

Other comments:

3) Can the author speculate as to why DFTB predicts the wrong behavior? Or point to possible explanations?

4) Were van der Waals functionals used to determine the equilibrium interlayer distance?

  • validity: good
  • significance: good
  • originality: good
  • clarity: high
  • formatting: excellent
  • grammar: good

Author:  Lorenzo Sponza  on 2022-11-16  [id 3032]

(in reply to Report 3 on 2022-11-07)
Category:
answer to question

We acknowledge receipt of the report of Referee 3 and we thank the referee for the positive advice on our work. The answers to the referee's questions are listed here below.

1) There are experimental observation that atomic displacements take place in hBN bilayers in order to maximise the size of domains with almost untwisted stacking, and this is confirmed by some simulations. We give some references in the main text. a) The reason why atomic rearrangements are expected to be weaker at larger twist angles is because locally the deviation from untwisted stacking is larger. As a consequence, the energy barrier to distort the structure and form untwisted domains is higher. Moreover, the energy gain is low since domains are small as a consequence of the short moiré periodicity. b) We haven't done any test on in-plane relaxation, but in Reference 53 [PRB 99, 205134] a set of three graphene moiré supercells of increasing size have been relaxed and it appears that the ratio between the size of untwisted domains and that of the intermediate stacking regions gets bigger for larger moiré cells (i.e. smaller twist angles). c) There is no 'a priori' reason to neglect in-plane relaxation, but, as explained above, there are reasons to think that this is less important at large twist angles. Actually, the choice of neglecting them is dictated by the objective of the study. We wanted to study the effect of stacking and twist angle on the band-gap, so we found it was wiser to start focusing on perfect structures to have a first 'clean' understanding of the effects. We can imagine to add in-plane relaxation later and compare what changes are introduced by the atomic rearrangement. Putting all ingredients from the very beginning would have made the effects much harder to unravel even though closer to experiment.

2) The question asked by the referee is fascinating and we thank the referee for having asked it. It forced us to go deeper in our formalism and derive a demonstration added to the new version of the manuscript as an Appendix that we hope is satisfactory. We show that all twist angles resulting in periodic moiré patterns, so all commensurate twisted bilayers, are included in our formalism. Basically, the key observation is that what really counts is the ratio 0<q/p<1. One can express the twist angle as a function of this ratio. The reasoning consists in the fact that you can establish a bijection between an angle comprises between 0 and 60˚ and a number a 0<x<1. You can always find a pair (q,p) corresponding to angles associated to rational x. For the irrational ones, of course you can’t find such a pair, but these angles correspond to incommensurate stackings which have no moiré periodicity.

3) This has been also asked by another referee. We added a sentence to the new version to answer this question. We think that TB parameters have not been chosen accurately enough.

4) Yes, we used PBE+van der Waals in the Grimme-d2 flavour. Another referee made us realise that we have never specified the exchange-correlation potential used in our calculations, so we added a sentence on the new version of the manuscript.

---

## Editorial Decision

resubmitted